There are amendments to this paper

# Atomically ordered non-precious $Co_3Ta$ intermetallic nanoparticles as high-performance catalysts for hydrazine electrooxidation

Guang Feng[1], Li An[1], Biao Li [1], Yuxuan Zuo[1], Jin Song[1], Fanghua Ning[1], Ning Jiang[1], Xiaopeng Cheng[2], Yuefei Zhang [2] & Dingguo Xia [1,3]*

Nano-ordered intermetallic compounds have generated great interest in fuel cell applications. However, the synthesis of non-preciousearly transition metal intermetallic nanoparticles remains a formidable challenge owing to the extremely oxyphilic nature and very negative reduction potentials. Here, we have successfully synthesized non-precious $Co_3Ta$ intermetallic nanoparticles, with uniform size of 5 nm. Atomic structural characterizations and X-ray absorption fine structure measurements confirm the atomically ordered intermetallic structure. As electrocatalysts for the hydrazine oxidation reaction, $Co_3Ta$ nanoparticles exhibit an onset potential of −0.086 V (vs. reversible hydrogen electrode) and two times higher specific activity relative to commercial Pt/C (+0.06 V), demonstrating the top-level performance among reported electrocatalysts. The Co-Ta bridge sites are identified as the location of the most active sites thanks to density functional theory calculations. The activation energy of the hydrogen dissociation step decreases significantly upon $N_2H_4$ adsorption on the Co-Ta bridge active sites, contributing to the significantly enhanced activity.

[1] Beijing Key Laboratory of Theory and Technology for Advanced Batteries Materials, College of Engineering, Peking University, Beijing 100871, P. R. China. [2] Institute of Microstructure and Property of Advanced Materials, Beijing University of Technology, Beijing 100124, P. R. China. [3] Beijing Innovation Center for Engineering Science and Advanced Technology, Peking University, Beijing 100871, P. R. China. *email: dgxia@pku.edu.cn

Highly efficient, low cost, and stable electrocatalysts are crucial for commercial applications of fuel cells. Considering the high cost, scarcity, and low operational stability of Pt catalysts, hindering the large-scale commercialization of fuel cell technology[1–4], some alternative catalysts based on non-precious metals have been investigated, including the use of transition metals[5,6], perovskites[7], carbon-based materials[8,9], and metal carbides[10,11]. These alternatives are from naturally abundant resources and are attractive due to their low cost and favorable catalytic performance. While these catalysts appear promising, their relatively low activity and poor long-term stability still cannot meet the requirements necessary for long-term use. Major challenges remain for research efforts focused on non-precious metal catalysts with high performance in fuel cells.

Compared with disordered alloys and monometallic nanocrystals, structurally ordered intermetallic nanomaterials can perform better as fuel cell electrocatalysts in terms of catalytic activity, long-term stability, and poison tolerance due to their definite composition, exceptional structural, and electronic properties[12–16]. Early investigations of ordered intermetallics as efficient fuel cell electrocatalysts focused on Pt-based late transition metal intermetallic nanoparticles (NPs) (e.g., PtCu[17–19], PtFe[20–22], PtCo[1,23–25], PtNi[26], and PtAg[27]), which reduced the platinum catalyst consumption and showed significant enhancement in catalytic activity and stability compared with Pt NPs. Recently, early transition metal (the group IIIB, IVB, and VB) intermetallic compounds have shown great potential as efficient fuel cell electrocatalysts[28–33]. For instance, DiSalvo and co-workers reported that atomically ordered $Pt_3Ti$ nanoparticles prepared with sodium naphthalide exhibited higher electrocatalytic current densities and much lower affinity for CO adsorption than atomically disordered $Pt_3Ti$, pure Pt, or Pt-Ru NPs for both formic acid and methanol oxidation reactions[28]. Hideki et al. found that both 150-nm $NbPt_3$ and 100-nm $TaPt_3$ intermetallic particles showed significant enhancement in catalytic activity and stability compared with Pt NPs[31,32]. Despite these obvious advantages, the works of the related early transition metal nano-intermetallic compounds are preliminary (e.g., $Pt_3Ti$[28,29], $Pt_3V$[29], $ZrPt_3$[30], $NbPt_3$[31], $TaPt_3$[32], and $Pt_3Y$[33]) due to higher melting points, more oxyphilic nature and much more negative reduction potentials of these metals compared with late transition metals. Furthermore, non-precious early transition metal nano-intermetallics, to the best of our knowledge, have not yet been reported.

Herein, we report an ordered $Co_3Ta$ intermetallic compound with an average particle size of 5 nm and uniform distribution on carbon supports. High-angle annular dark-field scanning transmission electron microscopy (HAADF-STEM) imaging and X-ray absorption fine structure (XAFS) measurements reveal the ordered intermetallic crystal structure of $Co_3Ta$. As an electrocatalyst for hydrazine oxidation reaction (HzOR), $Co_3Ta/C$ NPs exhibit outstanding performance for both activity and stability, including an ultralow onset potential ($E_{on}$) of −0.086 V (vs the reversible hydrogen electrode, RHE) and twofold improvement of specific activity relative to commercial Pt/C (+0.06V). To the best of our knowledge, the superior catalytic activity for HzOR is the top-level performance among the reported electrocatalysts. XAFS measurements and density functional theory (DFT) theoretical calculations identify that the Co-Ta bridge sites are the location of the most active sites of HzOR in the ordered $Co_3Ta$. The tuning of the electronic structure of the ordered $Co_3Ta$ leads to the superior electrocatalytic hydrazine oxidation activity. The excellent performances on HzOR provide a potential application for $Co_3Ta$ NPs to be an anode catalyst in direct hydrazine fuel cells.

## Results

### Synthesis and characterization of $Co_3Ta$ intermetallic NPs.
To prepare early transition metal intermetallic compounds, extremely strong reductants and high temperatures are usually adopted, which leads to significant agglomeration and unwanted particle growth[28,30–33]. With this in mind, we instead added surface treated carbon supports to the reaction solution before the co-reduction of metal salts (details can be found in the Methods section). Supplementary Fig. 1 shows the powder X-ray diffraction patterns of the precursors and products. After the precursors were treated at 400 °C for 3 h, the diffraction peaks were identified but appeared broad, allowing for assignment of $Co_3Ta$ as a face-centered cubic (fcc) structure in the Pm-3m space group (JCPDS, No. 15-0028). Broad diffraction peaks are indicative of small particles size. A representative HAADF-STEM overview image of the $Co_3Ta$ NPs is shown in Fig. 1a. The $Co_3Ta$ NPs are well dispersed on the carbon supports, with an average diameter of 5 nm, which is much smaller than that of other reported Pt-based early transition metal intermetallic compounds (Supplementary Table 1). The high-resolution transmission electron microscopy (HRTEM) image of $Co_3Ta$ NPs is shown in Fig. 1b, with lattice spacing is 2.10 Å corresponding to the (111) plane of intermetallic $Co_3Ta$. The same lattice spacing measured for other NPs (Supplementary Fig. 2) suggests that $Co_3Ta$ NPs have (111) basal planes. Energy-dispersive spectroscopy (EDS) mapping and line scanning profiles of a single $Co_3Ta$ NP are presented in Fig. 1c–e and Supplementary Fig. 3, respectively, revealing that both Co and Ta are homogeneously dispersed. Combined with the result of inductively coupled plasma (ICP) analysis, the molar ratio of these two elements is 75.95:24.05 (Fig. 1f), that is 3:1 ratio of $Co_3Ta$.

High-magnification HAADF-STEM imaging was employed to reveal the ordered intermetallic crystal structure of $Co_3Ta$. From the ordered arrangement of atoms, we can identify the ordered intermetallic structure of $Co_3Ta$. Figure 2a shows a representative atomic resolution image of $Co_3Ta$ along the [111] zone axis, with lattice spacing of 2.56 Å corresponding to the (110) plane. Because of the 'Z-contrast' of Ta and Co, the Ta columns display a higher intensity than that of the Co columns in HAADF-STEM images. It can be seen that each Ta atom is surrounded by six Co atoms in a periodic hexagonal array. The corresponding fast Fourier transformation (FFT) pattern (Fig. 2b) is matched with the simulated image in Fig. 2c, further demonstrating the crystallinity and ordered intermetallic phase of $Co_3Ta$. This ordered intermetallic structure also presents along the [211] zone axes. In Fig. 2d, a HAADF-STEM image of $Co_3Ta$ NP is viewed along the [211] zone axis with lattice spacings of 2.09 and 2.56 Å, which are assigned to (111) and (110) lattice fringes of the $Co_3Ta$ intermetallic structure. Due to the overlap of Ta and Co atoms in some positions along the [211] zone axis, the 'Z-contrast' of Ta and Co is small. The FFT pattern is shown in Fig. 2e. To further confirm the ordered structure, we have simulated the diffraction patterns of an ideal $Co_3Ta$ crystal along its [211] zone axis (Fig. 2f), which matches the experimental result. This is conclusive evidence that the atomically ordered $Co_3Ta$ intermetallic nanostructures have been successfully synthesized.

To further determine the entire ordered intermetallic structure of the $Co_3Ta$ NPs, we carried out X-ray absorption fine structure (XAFS) measurements for the $Co_3Ta$ NPs and other possible phases. Comparisons between the calculated and experimental absorption patterns of Co and Ta in the ordered intermetallic structure of $Co_3Ta$ and other potential samples (i.e., CoO, $Co_3O_4$, Co foil, $Ta_2O_5$, and Ta powder) are shown in Fig. 2g, h. The Computational Methods show how to obtain the theoretical spectrum. The experimental spectrum of the sample has been successfully reproduced, both for the peak positions and

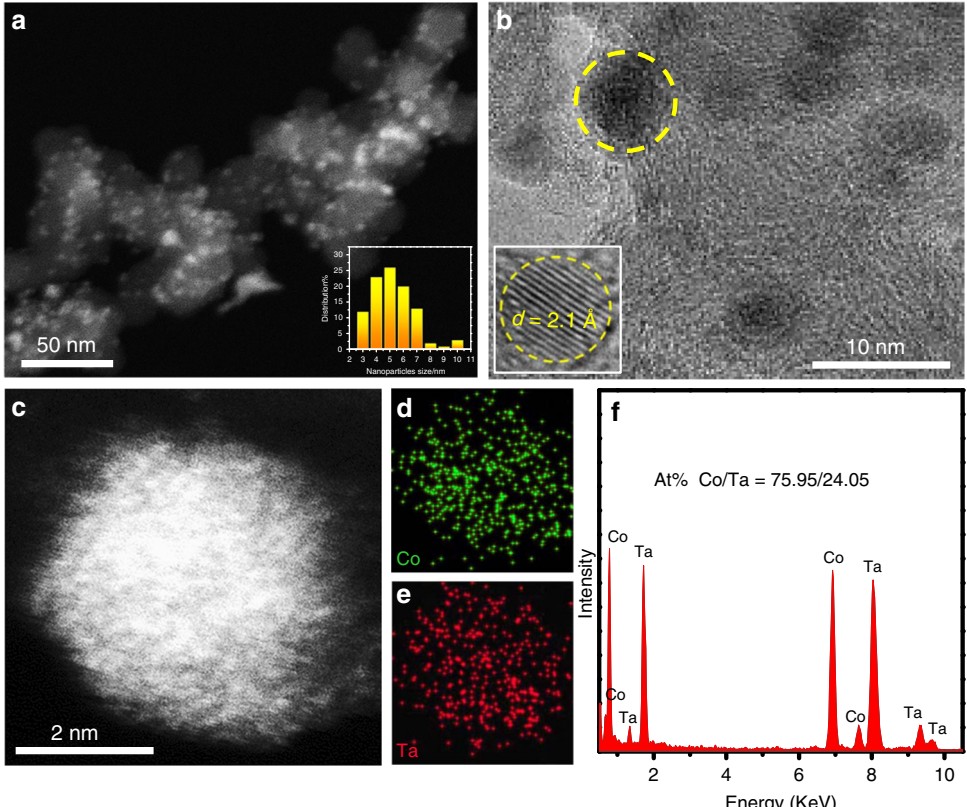

**Fig. 1** Characterization of the morphology and composition of Co$_3$Ta intermetallic nanoparticles. **a** Representative HAADF-STEM image and size distribution histogram (inset) of Co$_3$Ta intermetallic NPs. **b** The HRTEM image (inset shows the crystal lattice) of Co$_3$Ta intermetallic NPs. **c** HAADF-STEM image of a single NP. EDS mapping images of (**d**) Co and (**e**) Ta. **f** EDS spectrum of the Co$_3$Ta NP shown in (**c**)

intensities, by using the theoretical Co$_3$Ta configuration. Using the "fingerprint" of the Co K-edge and Ta L$_3$-edge XAFS, we can easily determine the dominant existence of the ordered intermetallic Co$_3$Ta phase structure. As to the small difference observed in the A peak site (Fig. 2g), this may be attributed to oxidation of the Co$_3$Ta surface upon exposure of sample in air for relatively long period during the measurements. X-ray photo-electron spectroscopy (XPS) (Supplementary Fig. 4) also confirms the presence of a slight oxidation, which is in agreement with previous metal nanocrystal studies[3,34–36]. The XAFS results strongly suggest that the as-prepared product is an ordered intermetallic Co$_3$Ta phase structure as a whole.

**Electrochemical performance of Co$_3$Ta intermetallic NPs.** Direct hydrazine fuel cell (DHFC), as a clean energy for future transportation vehicles and portable devices, has drawn increasing attention in recent years due to its remarkably fascinating highlights[36–41]. For example, not only do DHFC has a higher energy density and theoretical voltage than those of hydrogen fuel cell and most direct liquid fuel cells, but also a much safer handling system than that of gas state of hydrogen fuel cell, and more environmentally friendly CO$_2$-free products than other direct liquid fuel cells. However, in previous studies, the onset potentials ($E_{on}$) of the reported electrocatalysts for HzOR remained very high, indicating that these electrocatalysts require much higher potentials (overpotentials) to start the HzOR and realize a certain degree of hydrazine oxidation. As a result, the practical energy density is reduced. Here, we found that the ordered intermetallic Co$_3$Ta has an ultrahigh intrinsic activity toward hydrazine electrooxidation in an alkaline medium, notably an ultralow $E_{on}$ of −0.086 V vs RHE.

To characterize the intrinsic activity of the Co$_3$Ta catalyst, the kinetic currents shown in this work are all normalized to the electrochemically active surface area (ECSA) for a quantitative and more convincing comparison. According to the previous reports[42–44], the ECSA was determined by measurement of the electrochemical double-layer capacitance (EDLC) (Supplementary Fig. 5). The electrocatalytic HzOR activities of Co$_3$Ta/C NPs measured in different concentrations of hydrazine with scan rates of 5 mV s$^{-1}$ are displayed in Fig. 3a. From the linear sweep voltammetry (LSV) curve for the solution without hydrazine, we can see that no obvious anodic current appears in the potential window. However, when placed in a 0.1 M hydrazine solution, a significantly rising anodic current appears. Furthermore, the current density increases with increasing hydrazine concentration, indicating that the Co$_3$Ta/C NPs are highly efficient for the HzOR.

In order to further evaluate the intrinsic superior catalytic performance of Co$_3$Ta/C NPs, several samples (including the precursors, Co/C, XC-72, commercial Ir/C, and commercial Pt/C) were employed as control samples in a solution containing 0.2 M hydrazine and 3 M KOH at a scan rate of 5 mV s$^{-1}$, as shown in Fig. 3b. The onset potential is usually regarded as a very important evaluation criterion to determine the catalytic performance of catalysts. It can be seen that the $E_{on}$ of Co$_3$Ta/C NPs is −0.086 V (vs RHE; i.e., −1.175 V vs SCE), which is 16, 35, and 141 mV lower than that of Co/C, commercial Pt/C, and commercial Ir/C, respectively (Supplementary Fig. 6). Compared with that of other catalysts reported in the recent literatures (Supplementary Table 2), the onset potential of the ordered intermetallic Co$_3$Ta/C catalyst is the lowest, indicating that Co$_3$Ta/C NPs have an ultrahigh intrinsic activity toward HzOR. In addition, the Tafel slope of Co$_3$Ta/C NPs (56.9 mV dec$^{-1}$) is

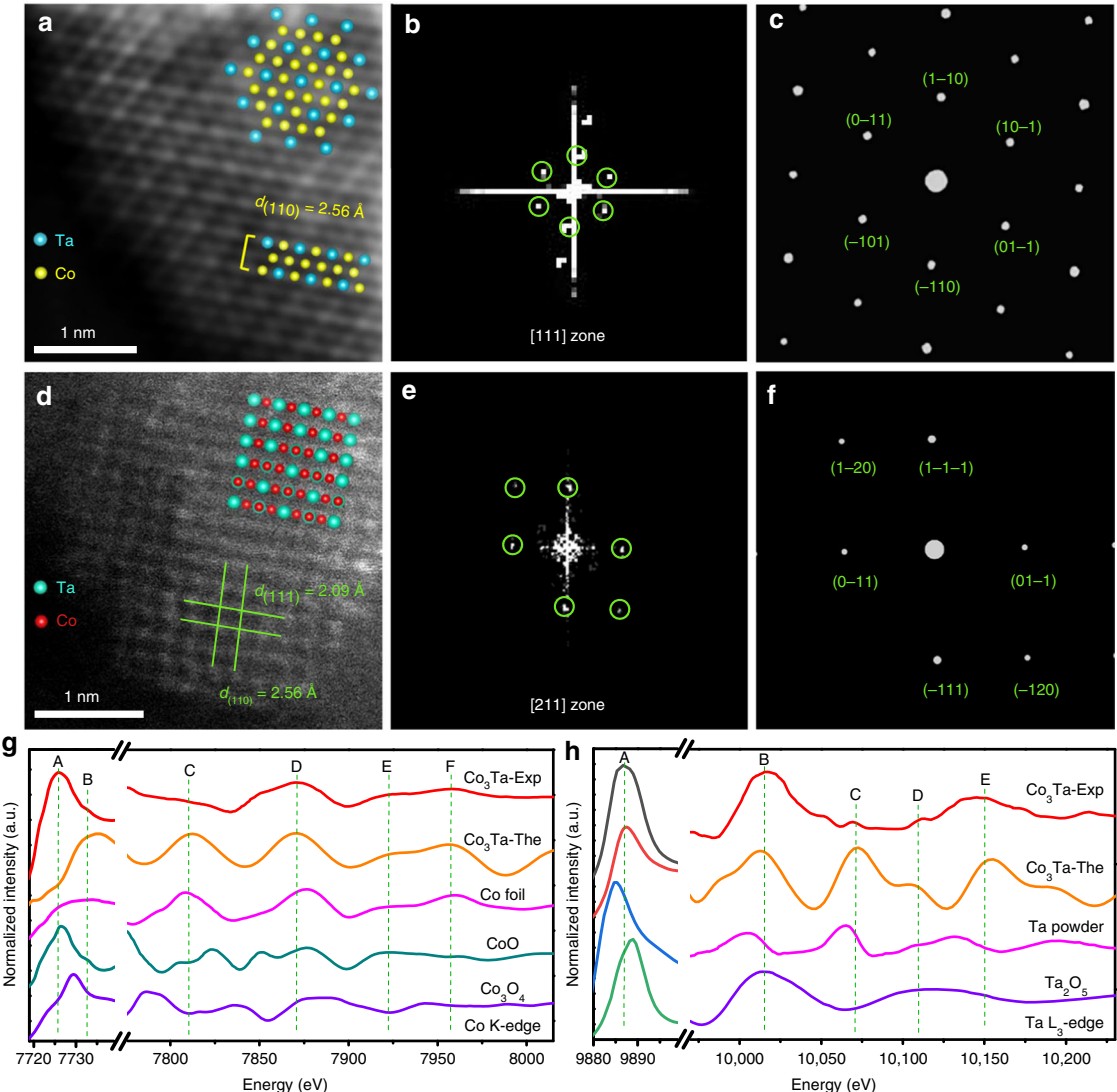

**Fig. 2** Atomic structural characterization and X-ray absorption fine structure spectrum measurement of $Co_3Ta$ intermetallic nanoparticles. **a** Atomically resolved HAADF-STEM image of $Co_3Ta$ intermetallic NPs. **b** The corresponding FFT of (**a**). **c** The simulated diffraction pattern from the [111] zone axis in (**a**). **d** Atomic HAADF-STEM image. **e** FFT pattern and corresponding simulated diffraction pattern (**f**) from the [211] zone axis in (**d**). **g** The absorption pattern comparisons at the Co K-edge of the experimental $Co_3Ta$ nanoparticles ($Co_3Ta$-Exp), calculated $Co_3Ta$ nanoparticles ($Co_3Ta$-Cal), Co foil, CoO, and $Co_3O_4$. **h** The absorption pattern comparisons at the Ta $L_3$-edge of the $Co_3Ta$-Exp, $Co_3Ta$-Cal, Ta powder, and $Ta_2O_5$

lower than those of Co/C NPs (60.6 mV $dec^{-1}$) and commercial Pt/C (73.7 mV $dec^{-1}$) (Fig. 3c), demonstrating that $Co_3Ta$/C NPs increase the current density more quickly than other samples during the HzOR. Notably, at a potential of +0.06 V (vs RHE) shown in Fig. 3d, $Co_3Ta$/C NPs produced a praiseworthy current (25.2 mA $cm^{-2}$), which is 1.67 and 2.02 times higher than that of Co/C NPs (15.1 mA $cm^{-2}$) and commercial Pt/C (12.5 mA $cm^{-2}$), respectively. Furthermore, compared with our previous studies (e.g., ultrathin nickel nanosheet arrays (68 mA $mg^{-1}$)[45] and ultrathin nickel–cobalt alloy nanosheet arrays (92 mA $mg^{-1}$)[46]), the ordered intermetallic $Co_3Ta$ shows a considerably high mass activity of 534 mA $mg^{-1}$ at a potential of +0.05 V (vs RHE) in 0.5 M hydrazine solution. The above results demonstrate that $Co_3Ta$/C NPs show ultrahigh electrocatalytic activity for the HzOR.

Stability is another important parameter to evaluate the practical performance of a catalyst. As shown in Supplementary Fig. 7, long-term stability of the $Co_3Ta$/C NPs for 12,000 s under a constant potential (+0.1 V vs RHE) in 0.2 M hydrazine solution was investigated. After this harsh 12,000 s test, $Co_3Ta$/C NPs

show a loss of only 12.8% of the initial current density (Fig. 3e). In contrast, Co/C NPs and commercial Pt/C exhibit a serious current density loss of 40.8% and 38.4%, respectively. Moreover, after 12,000 s, the $E_{on}$ of the $Co_3Ta$/C catalyst shows a positive shift of only 10 mV (Supplementary Fig. 8), electrochemical impedance spectroscopy (EIS) displays negligible changes (Supplementary Fig. 9), and the structure remains intact with no particle agglomerations (Supplementary Fig. 10). The above results demonstrate that $Co_3Ta$/C NPs are remarkably stable. The excellent stability and ultrahigh electrocatalytic activity of $Co_3Ta$/C NPs may be ascribed to its atomically ordered structure and electronic effect of $Co_3Ta$.

**X-ray absorption near-edge structure (XANES) analysis.** Figure 4a, b show the K-edge and $L_3$-edge X-ray absorption near-edge of Co and Ta elements in the ordered intermetallic structure of $Co_3Ta$ and other potential phases (i.e., bulk Co foil, CoO, $Co_3O_4$, bulk Ta powder, and $Ta_2O_5$). It can be seen that the Co K-edge and Ta $L_3$-edge X-ray absorption edge energy is arranged in

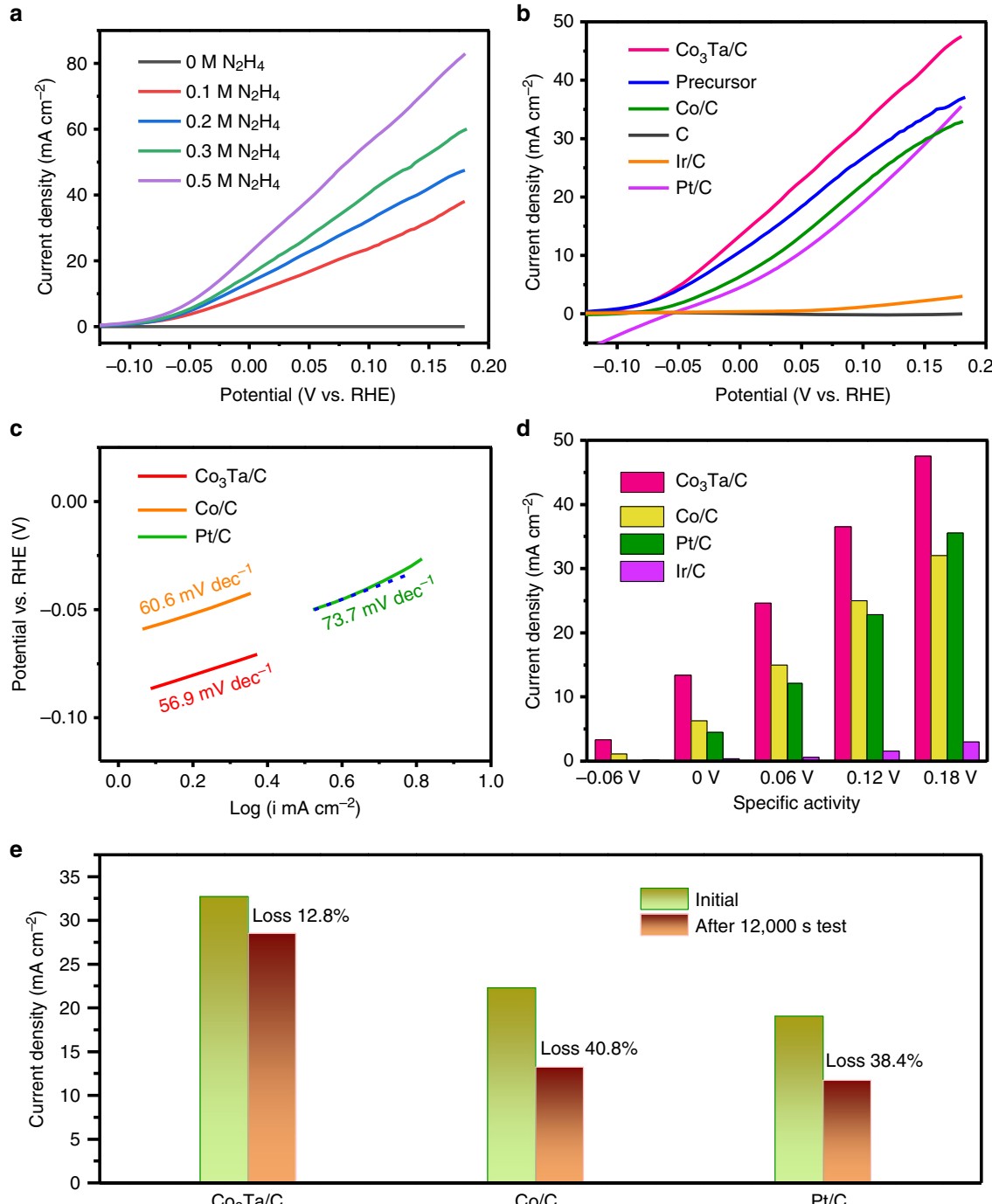

**Fig. 3** Electrochemical performance of $Co_3Ta/C$ electrocatalyst for hydrazine oxidation reaction. **a** HzOR polarization curves of $Co_3Ta/C$ intermetallic NPs at different hydrazine concentrations. **b** LSV curves of $Co_3Ta/C$, the precursors, Co/C, XC-72, commercial Ir/C, and commercial Pt/C in 3 M KOH solution with 0.2 M hydrazine. **c** The Tafel slopes of $Co_3Ta/C$ NPs, Co/C NPs, and commercial Pt/C; $Co_3Ta/C$ NPs quickly increase the current density for the HzOR. **d** HzOR performance comparisons of $Co_3Ta/C$, Co/C, commercial Pt/C, and commercial Ir/C at different potentials. **e** Quantitative comparison of the current densities of $Co_3Ta/C$ NPs, Co/C NPs, and commercial Pt/C before and after constant applied potential for 12,000 s; $Co_3Ta/C$ NPs are remarkably stable

the order of $Co_3O_4 > CoO > Co_3Ta \approx Co$ foil and $Ta_2O_5 > Co_3Ta \approx$ bulk Ta powder. The bader charge calculation[47,48] by using DFT method also confirms the slight charge transfer between Co and Ta in $Co_3Ta$, as shown in Supplementary Table 3. Despite the net gain of charge in the electron count from Ta due to the difference in electronegativity between Ta and Co, the Co absorption edge energy in $Co_3Ta$ exhibits no significant negative shift with respect to that of bulk metallic Co, which is

consistent with the previous works[49,50], and the Ta absorption edge energy in $Co_3Ta$ also exhibits a small positive shift with respect to that of bulk metallic Ta. Both are related with the size effect and the tuning of electronic structure of the Ta atom by Co, suggesting that the 5d hole in Ta atom increases[51,52]. An increase in the 5d hole will promote donation of electrons from the reactant orbital to the Ta 5d orbital. Thus, the tuning of electronic structure of Co and Ta likely facilitates the adsorption of $N_2H_4$ through

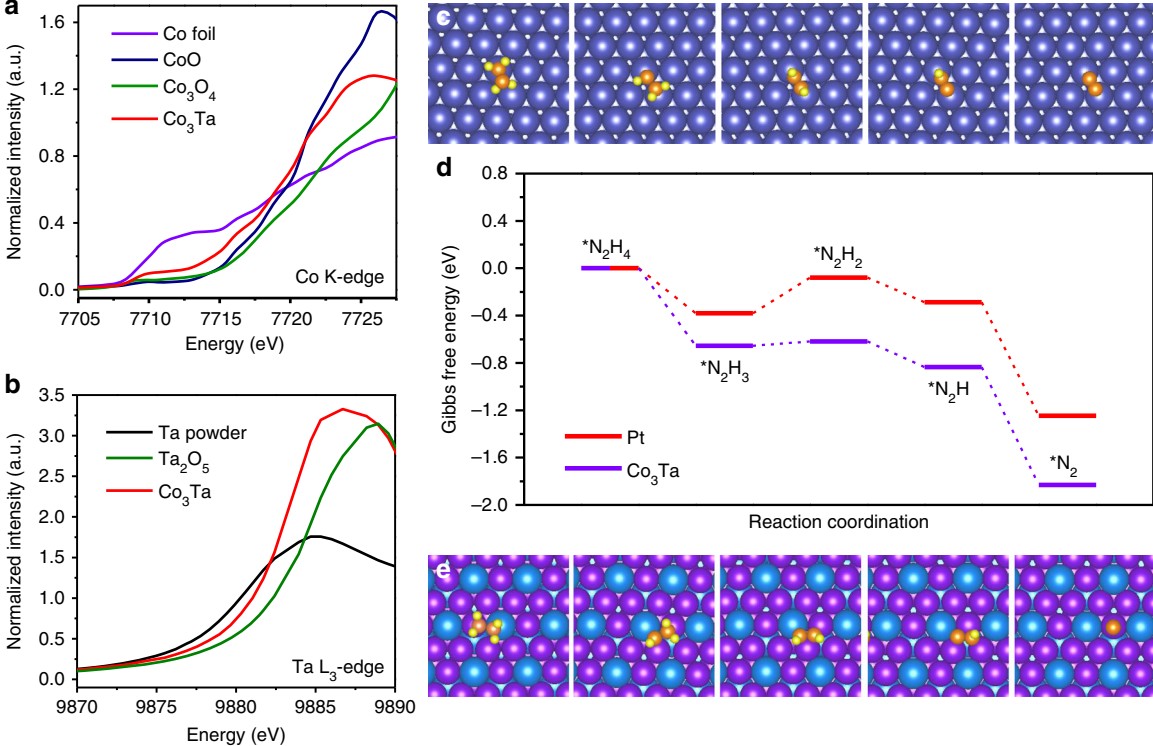

**Fig. 4** X-ray absorption near-edge structure analysis and theoretical calculations. **a** The normalized X-ray absorption near-edge at the Co K-edge of the $Co_3Ta/C$ NPs, Co foil, CoO, and $Co_3O_4$. **b** The normalized X-ray absorption near-edge at the Ta $L_3$-edge of the $Co_3Ta/C$ NPs, Ta powder, and $Ta_2O_5$. **c** A schematic illustration for the stepwise $N_2H_4$ molecular dehydrogenation process on the Pt (111) surface. Navy blue, orange, and yellow balls represent Pt, N, and H atoms, respectively. **d** Free energy profiles of the HzOR on the Pt (111) and $Co_3Ta$ (111) surfaces. **e** A schematic illustration for the stepwise $N_2H_4$ molecular dehydrogenation process on the $Co_3Ta$ (111) surface. Purple and cyan balls represent Co and Ta atoms, respectively

interaction of the lone pair electrons, weakening and accelerating breakage of the N–H bond, and increasing the hydrazine oxidation rate, which in turn increases the oxidation current of the anode, and ultimately enhances the electrocatalytic activity.

**Density functional theory (DFT) calculations**. Identifying the location of the active sites in an intermetallic is conducive to the design of optimal catalysts. Generally, there are several forms of hydrazine adsorption on $Co_3Ta$ catalyst surface[53]. As shown in Supplementary Fig. 11, the Ta-Co-cis conformation has the lowest adsorption energy (Supplementary Table 4), indicating the most stable adsorption configuration. It is preferable for the N–N molecular axis projection onto the surface be parallel with the Ta-Co bridge. That means the Co-Ta bridge sites can be identified as the location of the most active sites of HzOR in the ordered $Co_3Ta$ intermetallic.

DFT calculations were carried out to further reveal the origin of the superior intrinsic activity of ordered intermetallic $Co_3Ta$ toward hydrazine oxidation. The surface structure plays an important role about the computations of adsorption of hydrazine on metal surface[54]. Based on the HRTEM, XRD results, and the analysis of surface energies (Supplementary Table 5), the dehydrogenation process of hydrazine was analyzed over the $Co_3Ta$ (111) surface, with the dehydrogenation process of hydrazine over the (111) surface of Pt used as a control, as shown in Supplementary Fig. 12. Because both the Ta-Co-cis and Pt-anti conformations have the lowest adsorption energies (Supplementary Tables 4, 6), these conformations were selected as the adsorption configurations for DFT calculations. The density of states (DOS) of the $Co_3Ta$ (111) surface indicated that the electron structure of the surface is similar to bulk metal, and the surface DOS distribution crosses over the Fermi level, which

benefits electron transport during the HzOR process (Supplementary Figs. 13 and 14). Because of the magnetic properties of Co, the spin down parts are larger than the spin up parts near the Fermi energy level. Figure 4c, e illustrate the stepwise $N_2H_4$ molecular dehydrogenation process on the Pt (111) and the $Co_3Ta$ (111) surface, respectively, which consist of four intramolecular dehydrogenation steps $(N_2H_4 \rightarrow N_2H_3 \rightarrow N_2H_2 \rightarrow N_2H \rightarrow N_2)$[41,55–58]. Figure 4d shows the free energy profiles of the HzOR on the Pt (111) and $Co_3Ta$ (111) surfaces accordingly. The first dehydrogenation step $(N_2H_4 \rightarrow N_2H_3)$ is exothermic on the $Co_3Ta$ surface (0.65 eV), which is much higher than on the Pt surface (0.37 eV). This demonstrates that the first dehydrogenation step is more readily carried out on the surface of $Co_3Ta$ than that of Pt. Though the second dehydrogenation step $(N_2H_3 \rightarrow N_2H_2)$ is endothermic on both surfaces, the activation energy is much easier to overcome on the $Co_3Ta$ surface (0.03 eV) than on the Pt surface (0.31 eV). Overall, from the state of adsorbed $N_2H_4$ to the state of adsorbed $N_2$, the exothermic energy on the $Co_3Ta$ surface (1.83 eV) is much larger than that on the Pt surface (1.25 eV), indicating a more thermodynamically favorable catalytic process over the ordered intermetallic $Co_3Ta$.

Based on the discussion above, ordered intermetallic $Co_3Ta$ exhibits ultrahigh electrocatalytic activity toward hydrazine electrooxidation, including an ultralow onset potential, low Tafel slope, and high current density. The origin of the ultrahigh hydrazine oxidation activity can be explained as follows. First, the electronic structure of the Ta atom is tuned by Co and there is a synergistic effect between Co and Ta in $Co_3Ta$. Second, the activation energy of the hydrogen dissociation step decreases significantly during HzOR. Both will intrinsically contribute to the superior electrocatalytic activity of non-precious ordered nano-intermetallic $Co_3Ta$ toward hydrazine oxidation.

## Discussion

In summary, we have demonstrated the excellent performance and stability of novel $Co_3Ta$ NPs as an intermetallic electrocatalyst that contains non-precious metals and only early transition metals. The use of air-free synthetic conditions and subsequent annealing leads to the formation of atomically ordered $Co_3Ta$ NPs with a uniform particle size of 5 nm. As an electrocatalyst for hydrazine oxidation reaction (HzOR), $Co_3Ta/C$ NPs exhibit high stability and a higher electrocatalytic performance than conventional electrocatalysts in terms of their low onset potentials ($-0.086$ V vs RHE) for fuel oxidation. Theoretical calculations reveal that the activation energy of hydrogen dissociation decreases significantly upon $N_2H_4$ adsorption on the Co-Ta bridge active sites, which not only increases the number but also enhances the activity of the active sites, contributing to the considerably enhanced HzOR activity. The extraordinarily high performance of the supported ordered $Co_3Ta$ intermetallic nanocrystals provides a very promising alternative to the conventional Pt/C catalyst for the HzOR in direct liquid fuel cells.

## Methods

**Synthesis of $Co_3Ta$ intermetallic nanoparticles**. $Co_3Ta$ intermetallic NPs were synthesized by co-reduction of tantalum and cobalt salts under anhydrous and anaerobic conditions, followed by annealing. In a typical procedure, 0.18 mmol $CoCl_2$ and 0.065 mmol $TaCl_5$ were dissolved in 35 mL of rigorously dried and degassed tetrahydrofuran (THF) and diglyme solution in an argon-filled glovebox (both $O_2$ and $H_2O$ concentrations were <0.1 ppm), and stirred to form a clear solution. Then 60 mg of treated XC-72 carbon powders were added to the clear solution, and the mixture was stirred continuously for 8 h to form a uniformly dispersed solution. Next, sodium triethylborohydride (NaEt$_3$BH, 1 M in THF, Sigma-Aldrich) was injected into the mixture under vigorous stirring and left to stir overnight. Finally, the sample was separated from the above mixture via centrifugation without contacting air. The sample was then washed with rigorously dried and degassed THF and hexanes, and dried at 60 °C for 8 h in a glovebox. The obtained product was quickly transferred to a tube furnace and treated under flowing $H_2$/Ar at 300 °C for 3 h. The sample was then washed with argon-saturated ultrapure water and dried at 80 °C for 8 h in a vacuum oven. The precursor was again transferred to tube furnace and annealed at 400 °C for 3 h. The obtained intermetallic $Co_3Ta$ NPs were finally stored in a glovebox under Ar for further characterization and electrochemical measurements. The synthesis of Co/C NPs followed a similar procedure to that of $Co_3Ta$ NPs, except the addition of $TaCl_5$.

**Material characterizations**. The morphologies and composition of samples were characterized by high-resolution transmission electron microscopy (HRTEM; JEOL, JEM-2100F, 200 kV) equipped with an energy-dispersive X-ray spectrometry (EDS) instrument. Atomic structural characterization of the samples was measured using a spherical aberration corrected transmission electron microscope (TEM; Titan-G2, 300 kV). During the TEM measurements, electron exposures employed should be very low to minimize irradiation damage. X-ray diffraction (XRD) patterns were collected on an X-ray diffractometer (Bruker D8, Cu K$_\alpha$, $\lambda = 1.5406$ Å, 40 kV, and 40 mA) with a counting time of 8 s, recorded with $2\theta$ ranging from 15° to 70°. X-ray photoelectron spectroscopy (XPS) measurements were performed using a Thermo VG Scientific ESCALAB 250 spectrometer (Al K$_\alpha$, 200 W). The XAFS (Ta L$_3$-edge and Co K-edge) spectra were collected at beamline BL14W1 of the Shanghai Synchrotron Radiation Facility. It should be noted that the samples were kept under an argon atmosphere before all of the above characterizations to avoid oxidation.

**Electrochemical measurements**. All the electrochemical measurements of the HzOR were carried out in a standard three-electrode electrochemical cell at room temperature using a BioLogic SP 240 electrochemical workstation. The glassy carbon (GC) film, saturated calomel electrode (SCE), and $Co_3Ta/C$-coated GC rotating disk electrode (geometric area 0.1257 cm$^2$) were used as the counter electrode, reference electrode, and working electrode, respectively. The working electrode was prepared as follows: 3 mg of $Co_3Ta/C$ sample was dispersed in 1 mL of isopropyl alcohol and 5% Nafion solution, then the mixture was transferred to an ultrasonic bath to form a homogeneous catalyst ink. A 15 μL aliquot of the obtained suspension was pipetted onto the GC electrode and dried naturally for electrochemical measurements. The working electrode fabrication procedures of (20 wt%) and Ir/C (10 wt%) followed the same procedure with that of $Co_3Ta/C$ sample. Both Pt/C and Ir/C loadings are 0.36 mg cm$^{-2}$. The HzOR tests were conducted in 3 M KOH solution containing 0.5 M hydrazine at a rotation rate of 2000 rpm. In this work, the potentials measured (SCE) were converted to reversible hydrogen electrode (RHE) using the conversion method: $E_{RHE} = E_{SCE} + 0.242 + 0.059$ pH V.

**Computational methods**. All spin unrestricted DFT calculations were performed by using the Vienna Ab-initio Simulation Package (VASP)[59,60] with Perdew-Burk-Ernzerhof (PBE)[61] functional. The planewave basis (kinetic energy cutoff values ECUT = 520 eV) and projector-augmented wave (PAW) pseudopotential[62,63] were employed. The atomic positions were fully optimized until the Hellmann-Feynman force was <0.05 eV/Å and total energy convergence criterion was set to $1 \times 10^{-4}$ eV. The two Pt (111) and $Co_3Ta$ (111) surfaces were employed in our calculation using the 441 super cell, which includes four atomic layers, with the atomic position of the two top layers optimized, while the two bottom layers was fixed. The adsorption energies ($E_{ads}$) which one $N_2H_4$ molecule adsorbed on $Co_3Ta$ (111) surface with different layers were shown in Supplementary Table 7. The results indicated that four layers surface model in our calculation was workable. To avoid the interaction of neighboring images, ~15 Å vacuum layer was set to the direction of the $c$ axis. The Monkhorst-Pack[64] k-point grid $3 \times 3 \times 1$ was set in all calculations. The calculation of the Gibbs free energy of the intermediates followed the Nørskov method[65].

The calculation of the theoretical XAFs spectra of $Co_3Ta$: the calculation of the theoretical spectra was performed by the software Artemis based on a standard model of $Co_3Ta$ with a Pm-3m space group (JCPDS, No. 15-0028). The core of the calculation was based on the framework of FEFF's multiple scattering path expansion, where the simulated spectra is the summation of one or more scattering paths computed by FEFF. To obtain the theoretical spectra, the ATOM module was ran with a cluster size of 5.5 Å and longest scattering path of 5.0 Å. The calculation was done in the R space within an R range of 1.05–3.88 Å during which the Fourier transformation was conducted. Finally, the theoretical XAFS spectra were acquired by inverse Fourier transformation from R space and compared with experimental results.

The Gibbs free energy $\Delta G$ is defined as follows: $\Delta G = \Delta E + \Delta ZPE - T\Delta S + \Delta G_U + \Delta G_{pH}$, where $\Delta E$ is the change of electronic energy obtained from DFT calculations, ΔZPE is the change of zero-point energy, and $\Delta S$ is the entropy difference (see the values in Supplementary Table 8). $\Delta G_U$ is the free energy contributed by the electrode potential ($\Delta G_U = -n_e U$ ($n_e$ represents the number of electrons transferred in the corresponding elementary steps and $U$ is the electrode potential, respectively)). $\Delta G_{pH}$ is the free energy related to the H$^+$ concentration. The $\Delta G_U$ and $\Delta G_{pH}$ are set to zero in our calculation. The contribution of vibration of all adsorbed species were considered in our calculation.

The calculated Gibbs free energy change ($\Delta G$) is related to the onset potential in experiment. The theoretical $E_{onset}$ value could be obtained by the formula $E_{onset} =$ Max[$\Delta G_1$, $\Delta G_2$, $\Delta G_3$, $\Delta G_4$]/e. It should be noted that the theoretical $E_{onset}$ value obtained by the above formula are somewhat different from experimental value. This difference is caused by many factors involved in the real reaction conditions, such as solvent effect, coverage degree of catalytic species, experimental temperature, etc.

## Data availability

The data that support the findings of this study are available from the corresponding author upon reasonable request.

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

## Acknowledgements

This work was financially supported by the National Key R & D Program of China (No. 2016YFB0100200), the National Natural Science Foundation of China (No. 51671004 and U1764255), and the Beijing Municipal Natural Science Foundation (No. 2181001). The first-principles calculations were supported by High-performance Computing Platform of Peking University. All support for our work is gratefully acknowledged.

## Author contributions

G.F. and D.X. conceived the idea. G.F., L.A., and B.L. carried out the sample synthesis, characterization and performance measurement. L.A., F.N., and N.J. performed the DFT simulation and theoretical analyses. B.L., Y.Z., and J.S. helped with the XAFS measurements and discussion. X.C. and Y.Z. helped with the HAADF-STEM measurements and discussion. The paper was written by G.F. and D.X., and D.X. edited the paper. All authors contributed to discussing and revising the paper.

## Competing interests

The authors declare no competing interests.
