## [Peer Review File · Nature Communications]

Reviewers' comments:

Reviewer #1 (Remarks to the Author):

For the synthesis of non-precious early transition metal intermetallic nanoparticles, this is a novel work. Substantial evidence supports the successful preparation of intermetallic Co₃Ta and the top-level performance. The manuscript is well written and organized, but some questions need further explanation, especially the structure-activity relationship.

1. The authors are suggested to calculate the order degree of this Co₃Ta intermetallic nanoparticles.
2. How did the author get the theoretical XAFs spectrum of Co₃Ta?
3. The authors are suggested to state the loading of Pt/C and Ir/C.
4. As the authors mentioned, the condition of the synthesized Co₃Ta is rigorous (anhydrous and anaerobic conditions). How to determine the real active sites during the catalytic process in alkaline? (metal oxides or hydroxides in such a oxidation potential range)
5. In the discussion, the authors consider weakening and accelerating breakage of the N-H bond is closely related to the 5d hole in Ta atom. For Co₃Ta catalyst, Co is richer than Ta. What about the positive role of Co?

Reviewer #2 (Remarks to the Author):

In this work, authors reported the synthesis of nanoparticle Co₃Ta with grain size of 5nm. Then authors performed the test of hydrazine electrooxidation with Co₃Ta/C as the catalyst and found it is with good performance, compared with Pt/C. This point is very interesting. With DFT calculation, it is confirmed the surface of Co₃Ta is very active for the hydrazine. However, in the current state, I do not think the manuscript is ready for publication.

1. From the localized TEM, the crystal quality of Co₃Ta seems to be good. What is the reason that the signal of XRD is weak and the peak is very broad in Figure S1? We don't think it is due to the effect of grain sizes.
2. In Figure 3B, why does Pt/C have negative current under the onset potential, and others have no current?
3. For the surface of Co₃Ta nanoparticle, is it mostly from the contribution of (111) of bulk? Why is the (111) surface just considered in the calculation?
4. In Figure 4A and 4B, from the analysis of Co K-edge and Ta L₃-edge X-ray absorption edge energy, the charge transfer between Co and Ta seems to be ignored. It maybe perform DFT calculation to confirm this point.
5. How the Gibbs free energy is calculated? the contribution of vibration is considered?
6. For each reaction step, e.g. N₂H₄→N₂H₃, is there energy barrier for the reaction? If have, how about? Is it related to the onset potential?
7. Authors just consider one reaction chain N₂H₄→N₂H₃→N₂H₂→N₂H→N₂, is there other possible reaction chains?
8. In page 6, authors stated "The density of states (DOS) of the Co₃Ta (111) surface indicated that the electrical conductivity of the surface is similar to bulk metal (Figure S13), which benefits electron transport during the HzOR process". This statement is improper. The DOS can't indicate the electrical conductivity of the surface is similar to bulk metal. All the DOS at surface is non-zero for metals.
9. In Figure 3, the onset potential of precursor is similar to that of Co₃Ta/C. What is the precursor? why does precursor also have very low onset potential?
10. In the model of simulation, there is just 4 atomic layers. It seems to be too thin. Did authors consider the layer's thickness effect?
11. Is the Co₃Ta is ferromagnetic state or other state?

Responses to reviewer's Comments

Reviewer #1

For the synthesis of non-precious early transition metal intermetallic nanoparticles, this is a novel work. Substantial evidence supports the successful preparation of intermetallic Co₃Ta and the top-level performance. The manuscript is well written and organized, but some questions need further explanation, especially the structure-activity relationship.

Response: We greatly appreciate the reviewer's positive comments and valuable suggestions that help to improve the quality of our manuscript, and our responses have been listed below.

Comments 1. *The authors are suggested to calculate the order degree of this Co₃Ta intermetallic nanoparticles.*

Response: Thanks for your valuable suggestion. In order to confirm the crystallinity of the obtained products more comprehensively, we use various advanced characterization methods to characterize sample particles. As shown in Figure S2 in Supporting Information, the clear lattice fringes in different regions of the sample particles show that the samples have good crystal quality. The high-magnification HAADF-STEM images (Figure 2A and 2D) further confirmed the crystallinity and atomically ordered intermetallic phase of Co₃Ta. Moreover, we carried out XAFS measurements to determine the entire ordered intermetallic structure of the Co₃Ta NPs. Compared with the absorption patterns of the Co K-edge and Ta L₃-edge of the calculated ordered intermetallic Co₃Ta phase (Figure 2G and 2H), the as-prepared product can be easily determined as an ordered intermetallic Co₃Ta structure as a whole using the "fingerprint" of the Co K-edge and Ta L₃-edge XAFS.

Indeed, the ordered degree of alloy or intermetallic compounds is closely related to their electrocatalytic properties (*Nano Energy*, 2018, 50, 70-78; *Nano Lett.*, 2015, 15, 2468-2473). According to the reference (*Prog. Mater. Sci.*, 1995, 39, 159-241; *J. Alloy. Comp.*, 2018, 744, 791-

796), the order degree of the as-prepared Co_3Ta intermetallic nanoparticles was calculated to be about 0.906, which is consistent with that of intermetallic Ni_3Al (*Phys. Stat. Sol. (a)*, 1975, 32, 657-664). That means that the as-prepared Co_3Ta is highly ordered. This is also in good agreement with those of HAADF-STEM and XAFS mentioned above. The process of calculation is as follows:

After subtracting the XC-72 peak and background of X-ray diffraction pattern of the as-prepared Co_3Ta , the X-ray diffraction pattern of pure Co_3Ta is obtained (Figure R1). The ordered degree of intermetallic compounds is expressed by the long-range ordering parameter (LRO). The calculation formula is as follows:

$$\begin{aligned} \text{LRO} &= [(I^s/I_0^s)/(I^f/I_0^f)]^{1/2} \\ &= 0.906 \end{aligned}$$

Here, I^s and I^f represent the intensity of the diffraction peaks (110) of Co_3Ta superlattices and the strongest peaks of Co_3Ta , respectively. I_0^s and I_0^f represent the intensity of the diffraction peaks of the calculated Co_3Ta superlattices and the strongest (111) peaks of the calculated Co_3Ta , respectively.

Figure R1. X-ray diffraction pattern of the pure intermetallic Co_3Ta after subtracting the XC-72 peak and background.

Comments 2: *How did the author get the theoretical XAFs spectrum of Co₃Ta?*

Response: Thanks for your question. The calculation of the theoretical XAFS spectra was performed by the software Artemis based on a standard model of Co₃Ta with a Pm $\bar{3}$ m space group (JCPDS, No. 15-0028). The section of Computational Methods in the revised manuscript shows how to obtain the theoretical XAFs spectrum.

Changes made in the manuscript: In Computational Methods, we have added a brief description about the calculation of the theoretical XAFs spectrum of Co₃Ta, such as “The calculation of the theoretical spectra was performed by the software Artemis based on a standard model of Co₃Ta with a Pm $\bar{3}$ m space group (JCPDS, No. 15-0028). The core of the calculation was based on the framework of FEFF’s multiple scattering path expansion, where the simulated spectra is the summation of one or more scattering paths computed by FEFF. To obtain the theoretical spectra, the ATOM module was ran with a cluster size of 5.5 Å and longest scattering path of 5.0 Å. The calculation was done in the R space within an R range of 1.05-3.88 Å during which the Fourier transformation was conducted. Finally, the theoretical XAFS spectra were acquired by inverse Fourier transformation from R space and compared with experimental results”.

Comments 3. *The authors are suggested to state the loading of Pt/C and Ir/C.*

Response: Thanks for your kind suggestion. In this work, Commercial Pt/C (20 wt%) and Ir/C (10 wt%) were employed as control samples, and the kinetic currents are normalized to ECSA for a quantitative and more convincing comparison. The working electrode fabrication procedures of Pt/C and Ir/C followed the same procedure with that of Co₃Ta/C sample. Both Pt/C’s loading and Ir/C’s loading are 0.36 mg cm⁻².

Changes made in the manuscript: In the revised manuscript, we have added a sentence “The working electrode fabrication procedures of (20 wt%) and Ir/C (10 wt%) followed the same procedure with that of Co₃Ta/C sample. Both Pt/C’s loading and Ir/C’s loading are 0.36 mg cm⁻².”

Comments 4. *As the authors mentioned, the condition of the synthesized Co₃Ta is rigorous (anhydrous and anaerobic conditions). How to determine the real active sites during the catalytic process in alkaline? (metal oxides or hydroxides in such a oxidation potential range)*

Response: Thanks for referee's suggestion. Owing to the extreme hydrophilicity of reactants and very negative reduction potentials of tantalum, the synthesis of early-transition-metal (tantalum) alloy nanocrystals remains a formidable challenge, especially for that of atomically ordered intermetallic nanoparticles. The synthesis process of Co₃Ta intermetallic nanoparticles must be in an anhydrous and anaerobic condition. In this manuscript, XRD, TEM and EXAFS confirmed that as-obtained sample is ordered intermetallic Co₃Ta. We identified the phase of nanoparticles from the microscopic point of view by transmission electron microscopy, and found that the lattice fringe spacing of nanoparticles is in perfect agreement with the (111) crystal plane of ordered intermetallic Co₃Ta (JCPDS, No. 15-0028), and the atomic image arrangement is Co₃Ta configuration. From the macro-average information point of view, through the structural analysis of XAFS absorption spectra, we determined that the structure of Co₃Ta as a face-centered cubic (fcc) is in the Pm $\bar{3}$ m space group (JCPDS, No. 15-0028).

As reported in previous literatures (*ACS Catal.*, 2018, 8, 3237-3256; *J. Am. Chem. Soc.*, 2004, 126, 4043-4049; *Joule*, 2019, 3, 956-991), ordered intermetallic phases have definite composition and structure, the order enhances the chemical and structural stability of the catalytic particles. In other words, once intermetallic compounds have been successfully synthesized, they have considerable stability. This point can also be confirmed by other reported early-transition-metal intermetallic compounds ((e.g. Pt₃Ti (*J. Am. Chem. Soc.*, 2008, 130, 5452-5458), Pt₃V (*J. Am. Chem. Soc.*, 2014, 136, 10206-10209), Pt₃Ta (*Energy Environ. Sci.*, 2015, 8, 1685-1689), and ZrPt₃ (*ACS Appl Mater Interfaces*, 2014, 6, 16124-16130)). After harsh electrocatalytic conditions, these early-transition-metal intermetallic compounds can maintain initial composition and structure, possibly because the strong M1-M2 bonds inhibit surface segregation and/or leaching during the catalysis (*Energy Environ. Sci.*, 2015, 8, 1685-1689; *J. Am. Chem. Soc.*, 2008, 130, 5452-5458).

Here, it needs to be stated again that our synthesis was carried out in anhydrous and anaerobic glove box, and we used H₂/Ar reducing atmosphere during the calcination process. In the subsequent drying process, we also dried in anhydrous and anaerobic environment. So we could confirm that the synthesized material was Co₃Ta. Moreover, the environment during the HzOR measurement of sample is a nitrogen saturated and strongly reducible hydrazine hydrate solution. Moreover, we provide the Nyquist plots (Figure S9) and TEM morphology (Figure S10) after the measurement, which have little change compared with that of pre-reaction.

As to the real active sites, we have a clear description of the real active sites based on the reducibility test environment of hydrazine hydrate in the manuscript, as shown in Figure S11 and Table S4. We compared various adsorption modes of N₂H₄ molecules on Co₃Ta surface, and concluded that Ta-Co-cis is the most stable adsorption conformation based on DFT calculations and EXAFS measurements. Meanwhile, from the free energy profiles of the HzOR on the Pt (111) and Co₃Ta (111) surfaces, and a schematic illustration for the stepwise N₂H₄ molecular dehydrogenation process on the Co₃Ta (111) surface, as shown in Figure 4D and 4E, we can see that both Co and Ta are catalytic active sites, and there is a synergistic effect between Co and Ta.

Comments 5. *In the discussion, the authors consider weakening and accelerating breakage of the N-H bond is closely related to the 5d hole in Ta atom. For Co₃Ta catalyst, Co is richer than Ta. What about the positive role of Co?*

Response: Thanks for referee's question. Due to the difference in electronegativity between Ta and Co, the electronic structure of the Ta atom is tuned by Co. Figure 4A and 4B in the manuscript show the K-edge and L₃-edge X-ray absorption near-edge of Co and Ta elements in the ordered intermetallic structure of Co₃Ta and other potential phases, respectively. It can be seen that the Co K-edge and Ta L₃-edge X-ray absorption edge energy is arranged in the order of Co₃O₄ > CoO > Co₃Ta ≈ Co foil and Ta₂O₅ > Co₃Ta ≈ bulk Ta powder. Despite the net gain of charge in the electron count from Ta due to the difference in electronegativity between Ta and Co, the Co absorption edge energy in Co₃Ta exhibits no significant negative shift with respect to that of bulk metallic Co, which is consistent with

the previous work (Small, 2014, 10, 2662-2669), and the Ta absorption edge energy in Co₃Ta also exhibits a small positive shift with respect to that of bulk metallic Ta. Both of them are related with the size effect and the tuning of electronic structure of the Ta atom by Co, suggesting that the 5d hole in Ta atom increases. An increase in the 5d hole will promote donation of electrons from the reactant orbital to the Ta 5d orbital. Thus, the tuning of electronic structure of Co and Ta likely facilitates the adsorption of N₂H₄ through interaction of the lone pair electrons, weakening and accelerating breakage of the N-H bond, and increasing the hydrazine oxidation rate, which in turn increases the oxidation current of the anode, and ultimately enhances the electrocatalytic activity.

Furthermore, by comparing several possible adsorption configurations (including Ta-gauche, Ta-anti, Ta-Ta-cis, Co-gauche, Co-anti, Co-Co-cis, and Ta-Co-cis), Ta-Co-cis conformation is selected as the most stable adsorption configuration due to its lowest adsorption energy. It is preferable for the N-N molecular axis projection onto the surface be parallel with the Ta-Co bridge. That means the Co-Ta bridge sites can be identified as the location of the most active sites of HzOR in the ordered Co₃Ta intermetallic. The activation energy of hydrogen dissociation decreases significantly upon N₂H₄ adsorption on the Co-Ta bridge active sites, which not only increases the number but also enhances the activity of the active sites, contributing to the considerably enhanced HzOR activity.

In brief, the electronic structure of the Ta atom is tuned by Co and both Co and Ta are catalytic active sites. There is a synergistic effect between Co and Ta in the ordered intermetallic structure of Co₃Ta.

Reviewer #2

In this work, authors reported the synthesis of nanoparticle Co₃Ta with grain size of 5nm. Then authors performed the test of hydrazine electrooxidation with Co₃Ta/C as the catalyst and found it is with good performance, compared with Pt/C. This point is very interesting. With DFT calculation, it is confirmed the surface of Co₃Ta is very active for the hydrazine. However, in the current state, I do not think the manuscript is ready for publication.

Response: Thanks for your careful review for our manuscript. We greatly appreciate your helpful and constructive comments. The point by point responses have been listed below.

Comments 1. From the localized TEM, the crystal quality of Co₃Ta seems to be good. What is the reason that the signal of XRD is weak and the peak is very broad in Figure S1? We don't think it is due to the effect of grain sizes.

Response: Thanks for referee's valuable question. Owing to the extreme hydrophilicity of reactants and very negative reduction potentials of early transition metals, the successful synthesis of early-transition-metal intermetallic nanoparticles was very limited, especially for non-precious early-transition-metal nano-intermetallics. In this manuscript, we have successfully synthesized non-precious metal Co₃Ta intermetallic NPs for the first time, with uniform size dispersion of 5 nm. XRD, TEM and EXAFS confirmed that as-obtained sample is ordered intermetallic Co₃Ta. We identified the phase of nanoparticles from the microscopic point of view by transmission electron microscopy, and found that the lattice fringe spacing of nanoparticles is in perfect agreement with the (111) crystal plane of ordered intermetallic Co₃Ta (JCPDS, No. 15-0028), and the atomic image arrangement is ordered intermetallic Co₃Ta configuration. From the macro-average information point of view, through the structural analysis of XAFS absorption spectra, we determined that the structure of Co₃Ta as a face-centered cubic (fcc) is in the $Pm\bar{3}m$ space group (JCPDS, No. 15-0028).

In order to confirm the crystallinity of the obtained products more comprehensively, we use various advanced characterization methods to characterize sample particles. As shown the clear lattice fringes in different regions of the sample particles in Figure S2 in Supporting Information, the samples show good crystal quality. The high-magnification HAADF-STEM images (Figure 2A and 2D) further confirmed the crystallinity and atomically ordered intermetallic phase of Co₃Ta. Moreover, we carried out XAFS measurements to determine the entire ordered intermetallic structure of the Co₃Ta NPs. Compared with the absorption pattern of the Co K-edge and Ta L₃-edge of the calculated Co₃Ta nanoparticles (Figure 2G and 2H), the as-prepared product can be easily determined as an ordered

intermetallic Co_3Ta structure as a whole using the “fingerprint” of the Co K-edge and Ta L_3 -edge XAFS.

To confirm that the broadening of the peak is mainly due to the small particle size, the samples were calcined at different temperature. As shown in Figure R2, XRD patterns show that all the peaks can be attributed to the face-centered cubic structure Co_3Ta with $\text{Pm}\bar{3}\text{m}$ space group (JCPDS, No. 15-0028). No impurity peaks corresponding to the Ta or Co-related compounds were observed. The intensity of diffraction peak of Co_3Ta increases with the increase of calcination temperature. This is because the crystallinity of particles increases with the increase of temperature and the growth of particles. Moreover, if the wt% of Co_3Ta is very small, the diffraction peak of Co_3Ta may be weak. However, the theoretical wt% of Co_3Ta is about 26% in the $\text{Co}_3\text{Ta}/\text{C}$ catalyst, which was confirmed by the strong L_3 edge patterns of Co_3Ta (Figure 4A, B).

Therefore, we think the broad diffraction peaks of the as-prepared Co_3Ta in this work are mainly indicative of small particles size.

Figure R2. Powder XRD patterns of $\text{Co}_3\text{Ta}/\text{XC-72}$ annealed at different temperature.

Comments 2. In Figure 3B, why does Pt/C have negative current under the onset potential, and others have no current?

Response: Thanks very much for your question. In Figure 3B, Pt/C has negative current under the onset potential, which is because of the hydrogen evolution reaction (HER). This is consistent with the previous works (*Angew. Chem. Int. Ed.* 2018, 57, 7649-7653). While the lines of others are flat from -0.125 to -0.05 V, indicating that no HER takes place.

Comments 3. For the surface of Co₃Ta nanoparticle, is it mostly from the contribution of (111) of bulk? Why is the (111) surface just considered in the calculation?

Response: Thanks very much for this question. Indeed, we employed HRTEM to characterize different nanoparticles in different regions (Figure 1B and Figure S2). We found that their lattice spacings were all corresponding to the (111) plane of intermetallic Co₃Ta, suggesting that Co₃Ta NPs have (111) basal planes. Meanwhile, the strongest peak appearing at 43.2° in XRD pattern of Co₃Ta (Figure S1) is consistent with the (111) plane. Moreover, we calculated the surface energies of different miller index surfaces of ordered intermetallic Co₃Ta, and found that the (111) plane has the lowest surface energy (Table R1), indicating the (111) plane is the most stable one.

In brief, the (111) plane is the most exposed surface and the most stable surface. And the (111) surface was just considered in the calculation.

Table R1. The surface energies of different miller index surfaces of Co₃Ta.

Surface type	Surface Energy (eV/Å ²)
(100)	0.175
(110)	0.174
(111)	0.155
(210)	0.182
(211)	0.177

Changes made in the manuscript: In the revised manuscript, we have added a sentence “The surface structure plays an important role about the computations of adsorption of hydrazine on metal surface (*Applied Surface Science*, 2015, 327, 462-469). Based on the HRTEM, XRD results, and the analysis of surface energies (Table S5), the dehydrogenation process of hydrazine was analyzed over the Co_3Ta (111) surface, with the dehydrogenation process of hydrazine over the (111) surface of Pt used as a control, as shown in Figure S12”. The reference (*Applied Surface Science*, 2015, 327, 462-469) was added in revised manuscript as Ref. 54. Table R1 was also added in supporting information as Table S5.

Comments 4. *In Figure 4A and 4B, from the analysis of Co K-edge and Ta L3-edge X-ray absorption edge energy, the charge transfer between Co and Ta seems to be ignored. It maybe perform DFT calculation to confirm this point.*

Response: Thanks for the reviewer’s kind suggestion. In the revised manuscript, Figures 4A and 4B show the K-edge and L_3 -edge X-ray absorption near-edge of Co and Ta elements in the ordered intermetallic structure of Co_3Ta and other potential phases. Despite the net gain of charge in the electron count from Ta due to the difference in electronegativity between Ta and Co, the Co absorption edge energy in Co_3Ta exhibits no significant negative shift with respect to that of bulk metallic Co. This is consistent with the previous work (*Small*, 2014, 10, 2662-2669). The Ta absorption edge energy in Co_3Ta exhibits a small positive shift with respect to that of bulk metallic Ta.

Generally speaking, the properties of chemical compound and materials are described in charge transfer between atoms, but the atomic charge in molecules or solids are not physical observables and it is not defined by quantum mechanical theory. There are many different computational schemes about atomic charge in molecular and solid have been proposed and the bader charge method is most common in solid calculation. In order to determine the charge transfer between Co and Ta in Co_3Ta . The bader charge of this ordered intermetallic structure was calculated. Firstly, the total charge density of Co_3Ta bulk was calculated by using DFT method, then the bader charge software developed by

Henkelman group (<http://theory.cm.utexas.edu/henkelman/code/bader/>) was employed and the bader charge results was listed in Table R2. It was confirmed the charge transfer between Co and Ta in Co₃Ta. As shown in Table R2, Co gain the some electrons from Ta, showing the tuning of electronic structure of the Ta atom by Co. which is consistent with the analytical results of XAFS.

Table R2. The net charge of Ta and Co in Co₃Ta from Bader charge analysis.

Element	Net Charge (e/atom)
Ta	1.08
Co-I	-0.23
Co-II	-0.36
Co-III	-0.49

Changes made in the manuscript: We have added the sentence “The bader charge calculation (*J. Phys. Chem. C*, 2015, 119, 8738-9774; *Surf Sci*, 2014, 622, 1-8) by using DFT method also confirms the slight charge transfer between Co and Ta in Co₃Ta, as shown in table S3” in the revised manuscript. The references (*J. Phys. Chem. C*, 2015, 119, 8738-9774) and (*Surf Sci*, 2014, 622, 1-8) were added in revised manuscript as Ref. 47 and Ref. 48, respectively.

Comments 5. *How the Gibbs free energy is calculated? the contribution of vibration is considered?*

Response: This is a very exquisite question which manifests that the reviewer is so careful and responsible. The Gibbs free energy ΔG is defined as follows: $\Delta G = \Delta E + \Delta ZPE + T\Delta S + \Delta G_U + \Delta G_{pH}$, where ΔE is the change of electronic energy obtained from DFT calculations, ΔZPE is the change of zero-point energy, and ΔS is the entropy difference. ΔG_U is the free energy contributed by the electrode potential, ($\Delta G_U = -n_e U$ (n_e represents the number of electrons transferred in the corresponding elementary steps and U is the electrode potential, respectively)). ΔG_{pH} is the free energy related to the

H⁺ concentration. The contribution of vibration of all adsorbed species were considered in our calculation.

Changes made in the manuscript: We have added a brief description about the calculation of Gibbs free energy ΔG in the section of Computational Methods.

Comments 6. For each reaction step, e.g. $N_2H_4 \rightarrow N_2H_3$, is there energy barrier for the reaction? If have, how about? Is it related to the onset potential?

Response: Thanks for your question. Based on the transition state theory, the transition state exist in the elementary chemical reaction and the energy barrier could exist in potential energy surface which along the reaction coordination. For example, the energy barrier of the thermal dissociation of N_2H_4 on Ni-Fe alloy surfaces was found by Jia's and Wu's group (*J. Phys. Chem. C*, 2015, 119, 8763-8774). Compare to the thermal reaction, the Gibbs free energy ΔG is more important in electrochemistry reaction simulation, because it is directly related to the onset potential. Therefore, the Gibbs free energy ΔG was considered in our calculation. This computational strategy was also employed in the previous works about the electrochemistry catalysis of N_2H_4 (*Nat. Commun.*, 2018, 9, 4365; *Adv. Mater.*, 2017, 29, 1604080).

Comments 7. Authors jut consider one reaction chain $N_2H_4 \rightarrow N_2H_3 \rightarrow N_2H_2 \rightarrow N_2H \rightarrow N_2$, is there other possible reaction chains?

Response: Thanks for your question. As reported in previous literature (*J. Power Sources*, 2008, 182, 520-523), hydrazine electrooxidation reaction in an alkaline medium would follow the following reaction paths:

We detected the gases producing in hydrazine electrooxidation reaction. As shown in Figure R3, the gas produced is mainly nitrogen, with little ammonia. Therefore, we just consider the 4-electron reaction as our calculation path. As we have known, the 4-electron reaction follows the chain $\text{N}_2\text{H}_4 \rightarrow \text{N}_2\text{H}_3 \rightarrow \text{N}_2\text{H}_2 \rightarrow \text{N}_2\text{H} \rightarrow \text{N}_2$ in an alkaline medium according to the previous works (*Nat. Commun.*, 2018, 9, 4365; *Adv. Mater.*, 2017, 29, 1604080; *Catal. Today*, 2011, 165, 80-88; *Surf. Sci.*, 2015, 632, 118-125).

Figure R3. The mass spectrometry curves of the gases in hydrazine electrooxidation reaction.

Comments 8. In page 6, authors stated "The density of states (DOS) of the Co_3Ta (111) surface indicated that the electrical conductivity of the surface is similar to bulk metal (Figure S13), which benefits electron transport during the HzOR process". This statement is improper. The DOS can't indicate the electrical conductivity of the surface is similar to bulk metal. All the DOS at surface is non-zero for metals.

Response: Thank you very much for this suggestion. We calculated the density of states DOS of bulk and surface of Co_3Ta (Figure R4 and Figure S13). The density of states (DOS) of the Co_3Ta (111) surface indicated that the electron structure of the surface is similar to bulk metal, and the surface DOS distribution crosses over the Fermi level, which benefits electron transport during the HzOR process.

Changes made in the manuscript: We changed the original sentence to “The density of states (DOS) of the Co_3Ta (111) surface indicated that the electron structure of the surface is similar to bulk metal, and the surface DOS distribution crosses over the Fermi level, which benefits electron transport during the HzOR process (Figure S13 and S14)” and “First, the electronic structure of the Ta atom is tuned by Co and there is a synergistic effect between Co and Ta in Co_3Ta . Second, the activation energy of the hydrogen dissociation step decreases significantly during HzOR.” in the revised manuscript. Figure R4 was also added in the support information as Figure S14.

Figure R4. The DOS curves of Bulk Co_3Ta .

Comments 9. In Figure 3, the onset potential of precursor is similar to that of $\text{Co}_3\text{Ta}/\text{C}$. What is the precursor? why does precursor also have very low onset potential?

Response: Thanks for your question. In this work, the precursor was obtained after one calcination, and the final product $\text{Co}_3\text{Ta}/\text{C}$ was obtained after two calcination. Therefore, the crystallinity of the precursor is slightly worse than that of the final product $\text{Co}_3\text{Ta}/\text{C}$, as can be seen from Figure S1. The precursor and the final $\text{Co}_3\text{Ta}/\text{C}$ have exactly the same composition. At low potential, the rate of

reaction is slow, and the small difference in crystallinity has little effect on the catalytic reaction, so it shows the performance of the precursor is even comparable to that for the $\text{Co}_3\text{Ta}/\text{C}$ at low potentials. However, as the potential increases and the reaction intensifies, the weak structural differences will greatly affect the performance. As we mentioned in the introduction, “structurally ordered intermetallic nanomaterials can perform better as fuel cell electrocatalysts in terms of catalytic activity due to their exceptional structural and electronic properties”. This result also could be found in previous reports (*Nano Lett.*, 2015, 15, 2468-2473; *Nano Energy*, 2018, 50, 70-78; *Adv. Energy Mater.*, 2019, 9, 1803040).

Comments 10. *In the model of simulation, these is just 4 atomic layers. It seems to be too thin. Did authors consider the layer's thickness effect?*

Response: Thank you very much for this question. The slab structure is used to calculate surface properties in theoretical study of surface. In order to determine the thickness effect of the slab model, the surface energy change of different layers of Co_3Ta (111) was calculated. As shown as in Fig R5, the surface energy difference between 4 layers and 8 layers is very small, so the 4 layers model used in our calculation is workable and the layer's thickness effect could be ignored.

Figure R5. The relationship between the surface energy and number of layers.

Comments 11. *Is the Co₃Ta is ferromagnetic state or other state?*

Response: Thank you very much for this question. We carried out magnetic measurements of the Co₃Ta/C at room temperature. From the magnetic hysteresis loops of Co₃Ta/C (Figure R6), we can see that the saturation magnetization of Co₃Ta/C is 0.52 emu/g at 4 kOe, and the sample remains magnetization when the external magnetic field is removed. So Co₃Ta/C exhibits a ferromagnetic behavior, and the magnetic property is very weak.

Figure R6. The magnetic hysteresis loops of the Co₃Ta/C measured at room temperature.

Reviewers' comments:

Reviewer #1 (Remarks to the Author):

The authors have addressed all the issues. It can be considered to publish.

Reviewer #2 (Remarks to the Author):

The responses are mostly satisfactory. The following issues are suggested to be clearly addressed in the manuscript or the supporting information,

- (1) About comments 6: Is the change of calculated Gibbs free energy related to the onset potential in experiment? How about this relation?
- (2) About comments 5: About the calculation of Gibbs free energy, how about the values for each item in the formula, such as energy change, zero point energy change, and entropy change?
- (3) About comments 10: Surface energy is not a direct indicator for the layer's thickness effect on surface reaction. We suggest authors perform the calculations of the adsorption on surface from different thickness.

Responses to Reviewer's Comments

Reviewer #1

The authors have addressed all the issues. It can be considered to publish.

Response: We warmly thank the reviewer for recommending our work for publication in *Nature Communications*.

Reviewer #2

The responses are mostly satisfactory. The following issues are suggested to be clearly addressed in the manuscript or the supporting information,

Response: Thanks for your careful review for our manuscript. We greatly appreciate your valuable and constructive suggestions that help to improve the quality of our manuscript. The point by point responses have been listed below.

Comments (1). *About comments 6: Is the change of calculated Gibbs free energy related to the onset potential in experiment? How about this relation?*

Response: Thanks very much for your question. In theory, the calculated Gibbs free energy change (ΔG) of the hydrazine oxidation reaction (HzOR) steps is related to the onset potential in experiment, due to the fact that the ΔG can be affected by the potential for an electrochemical catalytic reaction at the electrode (*J. Phys. Chem. B*, 2004, 108, 17886-17892; *Electrocatalysis*, 2014, 5, 68-74). Generally speaking, for an electrochemical catalytic reaction, the ΔG of a reaction step can be calculated as $\Delta G = \Delta E + \Delta ZPE - T\Delta S + \Delta G_U + \Delta G_{pH}$. $\Delta G_U = -n_e U$, where U is the potential at the electrode, and n_e represents the number of electrons transferred in the corresponding elementary steps. Based on this equation, the ΔG of each reaction step of HzOR can be obtained at a given pH and potential.

The onset potential calculation of HzOR is similar to that of electrochemical oxygen evolution reaction (OER), because both HzOR and OER are oxidation reactions (HzOR can be simply considered as the process of hydrazine being oxidized to nitrogen, and OER can be simply considered as the process of water being oxidized to oxygen). So the onset potential calculation of HzOR can refer to that of OER.

The elementary steps of hydrazine oxidation reaction (HzOR) in alkaline conditions can be listed as:

In order to model the thermochemistry of the HzOR, it is more convenient to work at acidic conditions (*J. Am. Chem. Soc.*, 2013, 135, 13521-13530), the steps (a-d) are modified as:

The two schemes (reactions (a-d) and (1-4)) are equivalent from a thermodynamic perspective, which is similar to the OER (*J. Am. Chem. Soc.*, 2013, 135, 13521-13530).

The Gibbs free energy changes for steps (1-4) can be expressed as:

$$\Delta G_1 = \Delta G(\text{N}_2\text{H}_3) - eU + \Delta G_{\text{H}^+}(\text{pH}) \quad (5)$$

$$\Delta G_2 = \Delta G(\text{N}_2\text{H}_2) - \Delta G(\text{N}_2\text{H}_3) - eU + \Delta G_{\text{H}^+}(\text{pH}) \quad (6)$$

$$\Delta G_3 = \Delta G(\text{N}_2\text{H}) - \Delta G(\text{N}_2\text{H}_2) - eU + \Delta G_{\text{H}^+}(\text{pH}) \quad (7)$$

$$\Delta G_4 = \Delta G(\text{N}_2) - \Delta G(\text{N}_2\text{H}) - eU + \Delta G_{\text{H}^+}(\text{pH}) \quad (8)$$

And the above $\Delta G(\text{N}_2\text{H}_3)$, $\Delta G(\text{N}_2\text{H}_2)$, $\Delta G(\text{N}_2\text{H})$, $\Delta G(\text{N}_2)$ were defined as:

$$\Delta G(\text{N}_2\text{H}_3) = E(\text{N}_2\text{H}_4^*) - E(\text{N}_2\text{H}_3^*) - 0.5E(\text{H}_2) + (\Delta ZPE - T\Delta S) \quad (9)$$

$$\Delta G(\text{N}_2\text{H}_2) = E(\text{N}_2\text{H}_4^*) - E(\text{N}_2\text{H}_2^*) - E(\text{H}_2) + (\Delta ZPE - T\Delta S) \quad (10)$$

$$\Delta G(\text{N}_2\text{H}) = E(\text{N}_2\text{H}_4^*) - E(\text{N}_2\text{H}^*) - 1.5E(\text{H}_2) + (\Delta\text{ZPE} - T\Delta S) \quad (11)$$

$$\Delta G(\text{N}_2) = E(\text{N}_2\text{H}_4^*) - E(\text{N}_2^*) - 2E(\text{H}_2) + (\Delta\text{ZPE} - T\Delta S) \quad (12)$$

The theoretical onset potential is then defined as:

$$E_{\text{onset}} = \text{Max} [\Delta G_1, \Delta G_2, \Delta G_3, \Delta G_4] / e = \Delta G_{\text{max}} / e \quad (13)$$

ΔG_{max} is the step in which Gibbs free energy is the largest in the whole reaction, e is the charge of an electron. And it means all Gibbs free energies different ($\Delta G_1, \Delta G_2, \Delta G_3, \Delta G_4$) at the potential of E_{onset} less than zero and all reaction is exothermic reaction.

From the equation (13), we can see that the theoretical onset potential is related to the calculated Gibbs free energy change (ΔG). However, it should be noted that the theoretical E_{onset} value obtained by the above formula are somewhat different from experimental value. This difference is caused by many factors involved in the real reaction conditions, such as solvent effect, coverage degree of catalytic species, experimental temperature, etc. This differences between theoretical and experimental value are reported by previous literatures (*ChemCatChem*, 2011, 3, 1159-1165; *J. Am. Chem. Soc.*, 2013, 135, 13521-13530). Some reports about HzOR (*Nat. Commun.*, 2018, <https://doi.org/10.1038/s41467-018-06815-9>; *Adv. Mater.*, 2017, 29, 1604080) also show the differences.

Changes made in the manuscript: We have added the sentences “The calculated Gibbs free energy change (ΔG) is related to the onset potential in experiment. The theoretical E_{onset} value could be obtained by the formula $E_{\text{onset}} = \text{Max} [\Delta G_1, \Delta G_2, \Delta G_3, \Delta G_4] / e$. It should be noted that the theoretical E_{onset} value obtained by the above formula are somewhat different from experimental value. This difference is caused by many factors involved in the real reaction conditions, such as solvent effect, coverage degree of catalytic species, experimental temperature, etc.” in the section of Computational Methods of the revised manuscript.

Comments (2). About comments 5: About the calculation of Gibbs free energy, how about the values for each item in the formula, such as energy change, zero point energy change, and entropy change?

Response: This is a very exquisite question which manifests that the reviewer is so careful and responsible. The reaction steps of electrocatalysis of N₂H₄ were listed: (* means the catalyst):

In order to obtain the Gibbs energy that referred the N₂H₄ energy, the reactions were defined as followed:

The Gibbs free energy for the above steps (1-4) were defined as:

$$\Delta G_1 = G(\text{N}_2\text{H}_4^*) - G(\text{N}_2\text{H}_3^*) - 0.5G(\text{H}_2) = E(\text{N}_2\text{H}_4^*) - E(\text{N}_2\text{H}_3^*) - 0.5E(\text{H}_2) + (\Delta ZPE - T\Delta S)$$

$$\Delta G_2 = G(\text{N}_2\text{H}_4^*) - G(\text{N}_2\text{H}_2^*) - G(\text{H}_2) = E(\text{N}_2\text{H}_4^*) - E(\text{N}_2\text{H}_2^*) - E(\text{H}_2) + (\Delta ZPE - T\Delta S)$$

$$\Delta G_3 = G(\text{N}_2\text{H}_4^*) - G(\text{N}_2\text{H}^*) - 1.5G(\text{H}_2) = E(\text{N}_2\text{H}_4^*) - E(\text{N}_2\text{H}^*) - 1.5E(\text{H}_2) + (\Delta ZPE - T\Delta S)$$

$$\Delta G_4 = G(\text{N}_2\text{H}_4^*) - G(\text{N}_2^*) - 2G(\text{H}_2) = E(\text{N}_2\text{H}_4^*) - E(\text{N}_2^*) - 2E(\text{H}_2) + (\Delta ZPE - T\Delta S)$$

All calculation results (energy change (ΔE), zero point energy change (ΔZPE), and entropy change (ΔS)) were listed in Table R1:

Table R1. The results of ΔE , ΔZPE and $T\Delta S$ for Co₃Ta (111) and Pt (111).

Reaction steps	Surface	ΔE (eV)	ΔZPE (eV)	$T\Delta S$ (eV)
$\text{N}_2\text{H}_4^* \rightarrow \text{N}_2\text{H}_3^* + \text{H}^+ + \text{e}$	Co ₃ Ta(111)	-0.59	-0.12	-0.06
	Pt(111)	-0.31	-0.11	-0.04
$\text{N}_2\text{H}_4^* \rightarrow \text{N}_2\text{H}_2^* + 2\text{H}^+ + 2\text{e}$	Co ₃ Ta(111)	-0.17	-0.33	0.12
	Pt(111)	0.39	-0.32	0.14
$\text{N}_2\text{H}_4^* \rightarrow \text{N}_2\text{H}^* + 3\text{H}^+ + 3\text{e}$	Co ₃ Ta(111)	-0.01	-0.51	0.30
	Pt(111)	0.54	-0.52	0.32

N ₂ H ₄ *→N ₂ *+4H ⁺ +4e	Co ₃ Ta(111)	-0.60	-0.68	0.55
	Pt(111)	0.01	-0.70	0.55

Changes made in the manuscript: In the revised manuscript, we have added a sentence “see the values in Table S8”. Table R1 was also added in supporting information as Table S8.

Comments (3). *About comments 10: Surface energy is not a direct indicator for the layer's thickness effect on surface reaction. We suggest authors perform the calculations of the adsorption on surface from different thickness.*

Response: Thanks for your valuable suggestion. To estimate the effect of thickness on the calculated results, the adsorption energies (E_{ads}) which one N₂H₄ molecule adsorbed on Co₃Ta (111) surface with 2 layers, 4 layers, 6 layers and 8 layers were calculated, respectively. The adsorption energy (E_{ads}) is defined as $E_{ads} = E_{N_2H_4\&surface} - E_{N_2H_4} - E_{surface}$, where $E_{N_2H_4\&surface}$, $E_{N_2H_4}$ and $E_{surface}$ are the total energies of N₂H₄ adsorbed surface, N₂H₄ and clean surface, respectively. As shown in Table R2, the adsorption energy (E_{ads}) with 4 layers thickness is similar to that with 6 layers and 8 layers. The results indicate that four layers surface model in our calculation is workable.

Table R2. The relationship between the adsorption energy and number of layers.

Layers	E(N ₂ H ₄ &surface)/eV	E(surface)/eV	E(N ₂ H ₄)/eV	E _{ads} /eV
2_layers	-91.58984268	-60.62858533	-30.29083739	-0.67041996
4_layers	-159.0308869	-128.0005605	-30.29083739	-0.73948901
6_layers	-226.6693516	-195.6227054	-30.29083739	-0.75580885
8_layers	-294.2978521	-263.2682775	-30.29083739	-0.73873719

Changes made in the manuscript: In the revised manuscript, we have added a sentence “The adsorption energies (E_{ads}) which one N_2H_4 molecule adsorbed on Co_3Ta (111) surface with different layers were shown in Table S7. The results indicated that four layers surface model in our calculation was workable.” Table R2 was also added in supporting information as Table S7.

REVIEWERS' COMMENTS:

Reviewer #2 (Remarks to the Author):

Authors have addressed properly the issues raised by us. This interesting work may be published.

Responses to Reviewer's Comments

Reviewer #2

Authors have addressed properly the issues raised by us. This interesting work may be published.

Response: We greatly thank you for reviewing our manuscript and really appreciate your valuable suggestions and comments.